# The Quality of the Supervisor–Nurse Relationship and Its Influence on Nurses’ Job Satisfaction

**DOI:** 10.3390/healthcare9101388

**Published:** 2021-10-17

**Authors:** Nieves López-Ibort, Miguel Angel Cañete-Lairla, Ana Isabel Gil-Lacruz, Marta Gil-Lacruz, Teresa Antoñanzas-Lombarte

**Affiliations:** 1Miguel Servet University Hospital, 50009 Zaragoza, Spain; nlopezi@salud.aragon.es (N.L.-I.); tantonnanzasl@salud.aragon.es (T.A.-L.); 2Psychology and Sociology Department, Education Faculty, San Francisco Campus, University of Zaragoza, 50009 Zaragoza, Spain; mcanete@unizar.es; 3Business Department, School of Engineering and Architecture, Rïo Ebro Campus, University of Zaragoza, 50018 Zaragoza, Spain; anagil@unizar.es; 4Psychology and Sociology Department, Health Science Faculty, San Francisco Campus, University of Zaragoza, 50009 Zaragoza, Spain

**Keywords:** leadership, perceived organizational support, nurse job satisfaction, supervisor–workers relationship

## Abstract

Background: Leader–Member Exchange theory provides strategic information about how to improve the leader’s role and nurses’ satisfaction on healthcare organizations. Objectives: The main objective of this research was to study the quality of the supervisor–nurse relationship in relation to the nurses’ job satisfaction. This research also analyses how the relationship between Leader–Member exchanges and nurse job satisfaction could be moderated by other variables, such as nurse psychological empowerment, nurse-perceived organizational support and Leader–Leader Exchange. Methods: The sample comprises of 2541 registered nurses who work in public hospitals in the Autonomous Region of Aragon (Spain). Regression analyses were conducted. Results: The statistically significant results demonstrate the influence that the supervisor’s leadership exerts on the job satisfaction of the nurse. Conclusions: The moderating variables (Empowerment, Perceived Organizational Support and Leader–Leader relationship) play an important role explaining the job satisfaction of the nurse. Deepening in these relationships could help us implement precise strategies to improve the nurse organizational commitment and the quality of health care performance.

## 1. Introduction

As Haque [1] points outs, scientific literature has evidenced how transformational, authentic and appreciative leadership influences, in a positive way, the employee well-being, including job stress and life satisfaction. 

Models, such as attribution theory, also explain the important role of workers’ perceptions and expectations in understanding the leader influence on the employees’ health-care behaviors and attitudes [2]. Social exchange theory focuses on the perceived organizational support, evidences how employees’ expectations about leader rewards equity according to their efforts, will be an important source of workers’ motivation and job performance [3]. From these perspectives, a nurse leadership role could also be understood as a work model for these subordinates and as a responsibility role related to health care employees’ well-being promotion and organizational sustainability [1,4]. 

About the leader and subordinates mutual interactions, up to now, most research has focused on the importance of the horizontal communication. In this sense, Leader-Member social exchange emphasis on the importance of the two-way (dyadic) relationship between leaders and other organization members [5]. 

Positive Leader-Member social exchanges (LMX) produce tangible and intangible benefits to the organization [6]. Intangible benefits include fluency in communication with leaders [7] and relationships that are based on trust [8]. Significant tangibles refer to relevance in decision-making [9], improvement of organizational participation/empowerment [10], professional promotion opportunities [11], and higher salaries [12]. All these benefits foster a more agreeable work environment that facilitates greater job satisfaction [13]. The partnership between the leader and his work team is fair or equitable when the leader provides resources in a manner that is perceived by the employee as beneficial [5]. 

Under these premises it is directly extrapolated that the quality of the LMX is positively related to job satisfaction [14,15] and promotes greater wellbeing and more suitable work behaviors among health professionals [16,17]. 

Job satisfaction has been identified as a relevant organizational output in the relationship with the LMX [18]. Job satisfaction is associated with staff commitment and permanence, both of which are beneficial and desirable objectives for organizations [19]. In addition to the positive relationships with supervisors, there are several factors that influence the job satisfaction of the registered nurse, such as autonomy in performance; work stress; burnout; organizational commitment [15]; staff turnover; unit performance [20]; results of patient care [21,22,23]; and, patient satisfaction [24].

The scientific literature review revealed a positive relationship between LMX and job satisfaction, although there are still open questions about the nature of this positive link [25]. The main aim of this current work is to study if organizational variables moderate the LMX-job satisfaction interaction. Even LMX is a key determinant of nurse job satisfaction, there are other relevant factors. Our research looks for this set of secondary variables and we measure their weight on LMX. 

Against this background, empirical literature about health care organizations emphasis on the following variables which are contemplated as moderators of the interaction between LMX and job satisfaction [15,16,18,26,27,28,29,30,31]. 

**Psychological empowerment** (EMP) can be a factor verifiable in the improvement of organizational results [10]. Potential organizational empowerment structures are: access to information; support; resources needed for work performance; and opportunities to learn and grow [32]. One of the most important determinants is the quality of the relationship with the supervisor [10]. Positive relationships with superiors and receiving more responsibilities and resources [33] increase the perception of empowerment, and consequently, leads to employees feeling that their work is more meaningful [34]. Empowerment is also an important determinant of job satisfaction [35]. Having more control over working practices and the autonomy to make decisions about patient care (a dimension of empowerment) lead to higher levels of satisfaction among nurses [26]. Psychological empowerment is especially significant in working environments like the healthcare sector where recent budget cuts have affected workers’ morale [27,28,29]. 

**Perceived organizational support** (POS) refers to the opinions of employees regarding the extent to which they believe that their organization values their contributions at work and cares about their wellbeing [36]. When employees feel supported by the organization (when the POS is high), they have confidence in it and tend to respond with positive cooperative behavior in order to achieve organizational goals [37,38]. Adequate nursing support practices in health care management are reflected in higher levels of job satisfaction [30,31]. 

**Leader–Leader Exchange** (LLX) completes the organizational hierarchy: supervisors are the link between lower-level workers and senior managers. Leaders have relationships with their subordinates (LMX) as well as relationships with other leaders and their immediate superiors (Leader–Leader Exchange—LLX). The relationship between leaders and subordinates is therefore also dependent on the relationships established between the supervisors and their peers and/or superiors [13,39]. 

As scientific literature evidences, these three variables are related to other crucial organizational variables, such as job commitment [40], organizational performance [41], profitability [42], and sustainability [43]. They configure a complex set of socially responsible human resource management strategies (SRHRM) [44]. 

## 2. Aim

The main aim of this research is to analyse how the quality of the supervisor–nurse interpersonal relationship is positively related to the job satisfaction of the nurse, controlled by moderating effects of psychological empowerment, perceived organisational support and Leader–Leader Exchange. We also confirm that moderating effects are influenced by hospital size.

### Hypothesis

**Hypothesis** **1.**
*Moderator variables (EMP, POS and LLX) modify the relation between leader-member exchange and job satisfaction.*


**Hypothesis** **2.**
*The moderator variables effect differs by the hospital size.*


## 3. Material and Methods

### 3.1. Participants 

The study universe comprised all nurses and all supervisors who worked in the nine public hospitals of the Autonomous Regional Community of Aragon in Spain. As Table 1 shows the size range of the hospitals goes from 122 to 1290 beds. One of the characteristics of this universe is its diversity: rural areas, small urban neighborhoods, and central university urban hospital are all represented. 

Questionnaires in hard copy were allocated to the 3628 nurses and 202 supervisors. Questionnaires were distributed to the total population of nurses who were active at the time of the study on these nine public hospitals. About the selection criteria, this research takes into account nurses who had been in relationship with the supervisor at least one month, and the supervisors who had had at least one month in the position. Previous studies suggest that two weeks is the minimum period to establish a leader-subordinated relationship [6]. For three months, the research director supervised the data collection with monthly remainders. After collection and elimination of the invalid ones, the sample consisted of 2541 nurses (70.04% of the nurse universe), 192 supervisors (95.05% of the supervisor universe), and 2500 matched dyads.

### 3.2. Data Collection

This is a descriptive cross-sectional study. The research unit was defined as a nurse/supervisor relationship of at least one month. In the first step, the research team leader explained the project and involved to the nursing directors and supervisors of the nine hospitals. After several meetings, one or two nurses of each hospital were assigned as responsible for collecting the questionnaires of their organizations. Participant anonymity was guaranteed because participants were provided with a hard copy and an envelope that could be sealed without contact information. Besides participation being voluntary, nurses assigned were responsible to motivate their colleagues and collect the envelopes in the hospitals and nurse departments. 

The dependent variable was the job satisfaction of the nurse; it was measured by the adapted questionnaire designed by Font Roja [45] by Aranaz and Mira [46]. The items were assessed using a Likert scale of five points ranging from 1 (strongly disagree) to 5 (strongly agree). The questionnaire measures the job satisfaction of professionals working in a healthcare environment. The sample analyzed in the present work gave a Cronbach alpha of 0.743; 51.2% of the total variance of the scale was explained by six factors. 

The quality of the nurse-supervisor relationship, measured by the LMX-7, was the independent variable. The LMX-7 is a dyadic measurement model that incorporates the simultaneous perceptions of the nurse (LMX (m)) and the supervisor (LMX (l)). 

The quality of the nurse-supervisor relationship as perceived by the nurse (LMX (m)), was measured using the one-dimensional adapted LMX-7 questionnaire of Graen and Uhl-Bien [5]. The instrument is an adaptation of the original test based on LMX Theory [47]. It includes seven items and a Líkert scale with 5 response options from 1 (rarely) to 5 (frequently). It is the most used tool for the measurement of LMX quality and has the best psychometric properties of similar instruments [48]. In previous studies, the Cronbach alpha has ranged from 0.8 to 0.9 [5]. Although the one-dimensionality of the LMX concept has been questioned [49,50], Graen and Uhl-Bien [5] conclude that the LMX integrates multiple dimensions, which may be inferred into a single integral measure. For the Spanish case, this questionnaire has already been used and validated in the banking sector by De la Rosa and Carmona [45] with a Cronbach alpha of 0.925. 

The quality of the nurse–supervisor relationship, as perceived by the supervisor (LMX (l)), was also measured with the one-dimensional LMX-7 (Leader–Member Exchange) questionnaire developed by Graen and Uhl-Bien [5]. Following the recommendation of Plagis and Green [51] the seven items (with the exception of the last item, which implies a global evaluation of the relationship) are adapted to the supervisor’s assessment of the subordinate’s contribution to the relationship. The LMX (l) questionnaire is, in effect, a mirror image of the traditional LMX (m); as an example, if a question in the LMX (m) questionnaire is: *To what extent do you think your supervisor is capable of understanding your problems and needs?* in the LMX (l) questionnaire the question would be: *To what extent do you think your nurse is capable of understanding your problems and needs*? It is therefore possible to examine the way in which both members of the dyad evaluate their exchange relationship. The items are scored on a Likert scale with five response options ranging from 1 (rarely) to 5 (frequently). The Cronbach alpha was 0.920.

The following moderating variables were also included:

The nurse’s empowerment (EMP) was measured by an adapted version of the Spreitzer questionnaire [52]. The questionnaire has 13 items and was validated in the Spanish language by Jáimez [53]. It measures: autonomy, competence, impact and meaning of work. It contains three items for each of the four dimensions of empowerment, except for the dimension of autonomy that has four. Items are scored on a Likert scale of 5 points ranging from 1 (little) to 5 (a lot). The Cronbach alpha was 0.881.

The nurse’s perceived organisational support (POS) was measured with the Eisenberger questionnaire. This 17-item questionnaire is an abridged version of the Survey of Perceived Organizational Support [36]. It was validated in the Spanish language by Ortega (2003) [54]. The items are evaluated on a 7-point Likert scale ranging from 1 (strongly disagree) to 7 (strongly agree). The Cronbach alpha was 0.938.

The supervisor’s perception of the quality of the supervisor–superior relationship (LLX) was (like the LMX (m)), measured with the LMX-7, the adapted one-dimensional questionnaire designed by Graen and Uhl-Bien [5]. As in the previous cases of application of the LMX, the different positions of those who make up the dyadic relationship were taken into account. The Cronbach alpha was 0.947.

Data was taken on the socio-demographic characteristics of sex, age and educational level (expert diploma or advanced studies/official master’s or specialty/degree in medicine/another degree).

Data was further taken on other organizational variables that have been found to affect the LMX:−The time that the nurse had been working: (a) in their current job [49]; (b) in the current unit; and, (c) with the current supervisor [31]. The results of these three variables were measured in years, if they did not reach one year, measurement was in decimals, proportional to the months.−The contract of the nurse (full-time or part-time).−The number of nurses who report directly to each supervisor.−The hospital size: large with more than 501 beds (hospitals 4 and 8) and small with 500 beds or less.−The population in the area of the hospital: Zaragoza (hospitals 4, 6, 7 and 8) and others (hospitals 1, 2, 3, 5 and 9).

### 3.3. Data Analysis

With the data base ready, first statistical procedure was to analyze the descriptive results of the variables. The use of regression equations was the main inferential statistical technique.

The significance of the variables to the explanation of the job satisfaction of the nurse was analyzed using a multiple linear regression model with a step-by-step method which is able to identify the effect of each variable whilst avoiding the problem of multicollinearity. The method progressively introduces the significant variables, starting with the one that has the strongest relationship with satisfaction then continuing with the others, in order of importance. At each step, the significance of the equation is studied to avoid the introduction of variables related to those already included in the equation (collinearity), because this results in a model that represents the best possible regression equation. In addition, it orders the variables by their importance or magnitude in the relationship with job satisfaction.

After that, we research the statistical significance of the selected moderator variables (95% level) through a new regression equation. Each regression shows the interaction among each one-moderator variable (EMP, POS, LLX) and the independent variable LM(x).

## 4. Results

Most participants (91.3%) were women (*n* = 2329), working full-time (78.4%). Their age average was 44 years old (SD = 11). Due to this age, the average work experience was high: 19.6 years (SD = 11.3). Stability of the time working in the current hospital was shown in a 14.5-year average (SD = 11.9) and long-time working with the current supervisor, with an average of 3.9 years (SD = 5.4).

### 4.1. Determinants of Job Satisfaction

The importance of the quality of the supervisor-nurse relationship, LMX (m) on the job satisfaction of nurses is undeniable. However, it is not the only variable that influences satisfaction: empowerment (EMP), perceived organizational support (POS) and the quality of the supervisor–superior relationship (LLX) produce, by themselves or through the interactions among them, a greater or lesser level of satisfaction.

The analysis of the explanatory weight of the variables, for the total sample and by hospital size, was used to identify which variables affect job satisfaction through the moderation of the LMX (m). Table 2 shows the descriptive statistics and correlations among variables. The low value of POS indicates that nurses differentiate between the quality of their relationship with their supervisor and that of the hospital as an organization. All the variables show relatively high and significant correlations among themselves, with the exception of the quality of the relationship of the supervisor with her superior (LLX), in relation to the job satisfaction of the nurse.

Table 3 indicates that the most important determinant of job satisfaction is perceived organizational support (POS), explaining 21.6% of its variance. Together, POS and Empowerment explain 32.85 of the variance. The final model includes the LMX (m) variable, which explains 36.5% of the variance.

### 4.2. Variables Related to Sociodemografic Characteristics and Job Satisfaction

Table 4 shows that in 5 out of 13 of the explanatory variables there is a significant and inverse correlation: the greater the magnitude of the explanatory variable, the lower the job satisfaction of the nurse. As the last four explanatory variables depend on hospital size, the differences in satisfaction based on that hospital size were explored in greater detail.

Through a multiple linear regression analysis summarized in Table 5, it is confirmed that explanatory variables differ in accordance with hospital size: large hospitals (more than 501 beds) and small hospitals (up to 500 beds).

The differences between the two types of hospital are clear, but not especially strong. For small hospitals, the most important moderate effect is caused by empowerment, followed by LMX (m) and POS. For large hospitals, the order was empowerment, POS and LMX (m).

### 4.3. Moderating Variables and Job Satistaction

Moderating variables are third variables that affect the correlation between two other variables [55]. Moderation is accepted if the product of the predictor variable and the moderating variable is significant in a regression equation which includes the dependent variable, the predictor, the moderator and the product.

The relationship between satisfaction and LMX (m) was direct, the value for the correlation coefficient was 0.43 (*p* < 0.001), indicating that the higher the satisfaction, the higher the score of the LMX(m) variable. The relationship between these variables and the existence of moderating variables was analyzed through linear, simple and multiple regression equations.

The equation that allows the forecasting of satisfaction based on the LMX (m) score was:(1)Satisfaction=2.67+0.19 LMX(m)+e

Both the ANOVA for the regression, F _(1, 2539)_ = 572.63, *p* < 0.001, and the contrast statistic for the significance of the t _(2539)_ = 23.93 and *p* < 0.001, indicate high significance and good predictive power, explaining 18.4% of the variance of Satisfaction.

The following variables were proposed as moderators: empowerment, perceived organizational support (POS) and quality of the supervisor–superior relationship (LLX). The equations included the term of the product that indicates a moderation effect. The PROCESS tool for SPSS was used for the calculation; PROCESS is a macro that is integrated into the regression menu with moderation analysis options [56].

#### 4.3.1. Psychological Empowerment

In Equation (1), each of the proposed moderating variables and their products are included with the variable LMX (m), which was significant. The relationship between empowerment and satisfaction (*r* = 0.44 and *p* < 0.001) would suggest, a priori, a moderating effect.
(2)Satisfaction=1.74+0.22 LMX+0.30 EMP 0.03 (LMX EMP)+e

The proportion of variance explained increases significantly up to 28.5%, indicating the positive effect that empowerment has on satisfaction (t _2540_ = 6.86 and *p* < 0.001). In addition, the regression coefficient for the product (t _2540_ = −2.07 and *p* = 0.04) confirms, with a confidence level of 95%, that empowerment is a moderating variable in the existing relationship between LMX (m) and the job satisfaction of the nurse.

#### 4.3.2. Perceived Organizational Support (POS)

The correlation between POS and job satisfaction was very high (r = 0.46 and *p* < 0.001). The regression equation was:(3)Satisfaction=2.39+0.14 LMX+0.14 POS 0.004 (LMX POS)+e

The proportion of explained variance rises to 29.1%, with a very significant regression coefficient for the POS variable (t _2540_ = 5.50 and *p* < 0.001). However, the regression coefficient obtained for the product is not significant (t _2540_ = −0.60 and *p* = 0.55), it is not, therefore, a moderating variable.

#### 4.3.3. Leader-Leader Exchange (LLX)

Although the LLX variable was not significantly related to job satisfaction (*r* = 0.029 and *p* > 0.05), the LLX variable was responsible for indirect effects that were not included in the first equation.

When the LLX and its product are included with the LMX (m), the following equation is obtained:(4)Satisfaction=2.63+0.19 LMX+0.01 LLX 0.001 (LMX LLX)+e

The moderating effect is not significant for the LLX. The proportion of explained variance hardly increases (from 18.4% to 18.6%) and the regression coefficient for LLX is very low (t _2499_ = 0.67 and *p* = 0.51). The coefficient of the product variable was also not significant, t _2499_ = −0.36 and *p* > 0.05. As the introduction of LMX does not significantly improve the first result it can be excluded from the analysis.

Summarizing, the only moderating variable for the total sample was empowerment. Its negative coefficient indicates that it has an inverse moderate effect on the relationship between LMX (m) and the job satisfaction of the nurse.

### 4.4. Moderating Variables of Job Satisfaction in Relation to Hospital Size

The process was then repeated, contemplating the variables in relation to hospital size: large hospitals—more than 501 beds—and small hospitals—up to 500 beds. The initial regression between satisfaction and LMX (m) was:

Small hospitals:(5)Satisfaction=2.68+0.18 LMX(m)+e

Explained variance was 17.0% and there was a significant regression coefficient, t _1628_ = 18.25, *p* < 0.001.

Large Hospitals:(6)Satisfaction=2.62+0.21 LMX(m)+e

Explained variance was 20.1% and there was a significant regression coefficient, t _911_ = 15.19, *p* < 0.001.

Table 6 shows the explained variance and the regression coefficients of the product variables for the moderating variables. The LLX is discarded because it does not have a significant relationship with the LMX(m). Results indicate that, in order to explain the nurse empowerment and nurse-perceived organizational support, LMX(m) is more important in large hospitals than in small ones.

As Figure 1 shows, positive nurse perception about their supervisor interactions influence especially high levels of job satisfaction. However, nurse empowerment is an important moderator variable of the relation between the nurse–supervisor interaction and nurse job satisfaction. When empowerment is high, this influence is lower than otherwise.

As proven for the general sample, empowerment had a significant and negative moderating effect for large hospitals, but not for small hospitals.

## 5. Discussion

This work has examined the complex interaction between the job satisfaction of the nurse and the perceived quality of leadership in the hospital. Job satisfaction depends, to large extent, on perceived organizational support and the degree of empowerment [35]. However, this first assertion should consider other variables that influence this relationship, such as hospital size.

Hypothesis 1: Moderator variables (EMP, POS and LLX) modify the relation between the leader–member exchange and job satisfaction.

Our research confirms that empowerment has a greater predictive power for job satisfaction in large hospitals than in small hospitals. This may be due to organizational bureaucratization and institutional depersonalization. It can be argued that nurses require intervention strategies that make them feel like active agents with enough resources to commit to the organization; in large hospitals, when the nurse is empowered, the strength of the relationship between the quality of leadership of the supervisor and the nurse’s job satisfaction decreases.

According to the reciprocity norm, the theory of the organizational support explains how the perceived organizational support is related with positive outcomes in employees, as for example organizational commitment [36] in healthcare system [57] and with nurse samples [58].

Perceived organizational support could also be studied as an external coping resource and could rely on the belief that the leadership and organization take care of workers’ well-being [3]. In our research, as in other studies, perceived organizational support is related to workers’ self-efficacy, and in this case, psychological empowerment [59]. As other studies confirm, nurse leadership support related to ethical competence improves nurses’ job satisfaction [60]. At the same time, perceived organizational support facilitates nurse empowerment and problem-focused coping strategies [3]. This is especially true in the COVID-19 crisis. Organizational support and individual support are different paths, that strengthen nurses’ ethical competence [60].

Hypothesis 2: The moderator variable effect differs by the hospital size.

On the other hand, our results highlight that human relations play a special role in smaller hospitals; and the relationship of the nurse with the supervisor has a greater specific weight in the assessment of the nurse’s welfare than in large hospitals.

This research provides robust empirical evidence on how the quality of the relationship that the supervisor establishes with the nurse improves job satisfaction. For example, in the research of Poikkeus et al. [60] about nurses and nurse leaders in Finland, positive correlations were found among perceived and organizational support, job satisfaction and ethical safety and ethical competence. The same tendency was evidenced in other backgrounds, prioritizing the altruism component of servant healthcare leadership in the perceived support and job satisfaction of their subordinate healthcare servants [61].

Results support the position that the training of leaders who are capable of motivating their collaborators in the execution of projects [62] should be prioritized within the health services. Due its relation with employee motivation and corporate social responsibility, SRHRM practices could be applied in the selection process to socially responsible nurse leaders [44]. Supervisors should be aware of the importance of their relations with registered nurses. The training must be adapted to the organization (for example hospital size) and offer a framework for resolving complex problems related to nursing and healthcare [63,64]. Specific training is requested in order to support nurse leaders in learning and applying the SRHRM [44]. Responsible leaders show both the healthcare management commitment for sustainability and positive support for their internal costumers (e.g., subordinated nurses) and external costumers (population) [65].

### 5.1. Limitations

This study selects a narrow set of variables, other motivational, psychological and organizational factors could be taken into account. Moreover, research was contextualized in hospitals in Aragon (Spain). This is a public health system in a regional sample that requires new studies in other environments, as for example, private health systems, big cities, other countries and so on. Cultural norms could influence these results. We have also been focusing on the dyadic study of the supervisor nurse and subordinates, but other units of analysis could be applied, as for example, the patient perspective.

### 5.2. Future Research

Due to the important role of nurse leadership communication, further research should be centered on their main skills, for example altruism, and how they are perceived and influence their subordinates [61]. Future research could introduce different sets of moderating variables: type of unit/service; primary care centers; private versus public health systems; public health policies by autonomous community, etc. Work could also be undertaken on the profile of the supervisor in relation to leadership performance. Longitudinal data reading, contemplating the leadership of the supervisor as a process that evolves over time might also be worth attention. This management issue is open to other ethical questions related to organizational social responsibility and societal needs and challenges (as COVID-19) [1,66].

## Figures and Tables

**Figure 1 healthcare-09-01388-f001:**
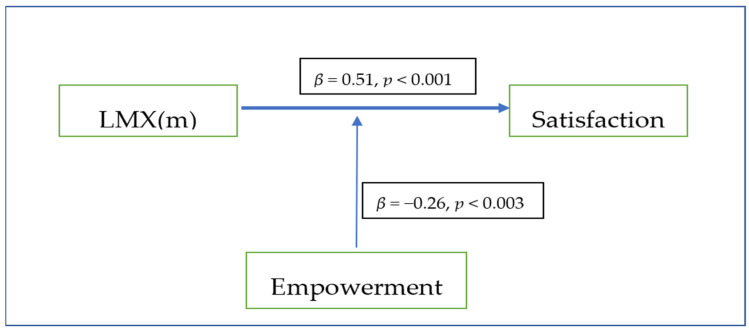
The hypothetical model with standardized parameters.

**Table 1 healthcare-09-01388-t001:** The hospitals where the study was conducted.

Hospital	N° Beds	Area Population
1. Alcañiz	125	16,043
2. Barbastro	161	16,961
3. Ernest Lluch	122	20,191
4. Lozano Blesa University Clinic Hospital	808	661,108
5. San Jorge	297	52,282
6. Nuestra Señora de Gracia	151	661,108
7. Royo Villanova	260	661,108
8. Miguel Servet University Hospital	1290	661,108
9. Obispo Polanco	215	35,564

**Table 2 healthcare-09-01388-t002:** Descriptive statistics and correlations.

Variables	1	2	3	4	5	M	SD
1. Satisfaction		0.429 **	0.445 **	0.463 **	0.029	3.32	0.36
2. LMX (m)	0.429 **		0.344 **	0.372 **	0.041 *	3.54	0.83
3. Empowerment	0.445 **	0.344 **		0.270 **	0.021	3.68	0.55
4. POS	0.463 **	0.372 **	0.270 **			3.33	0.99
5. LLX	0.029	0.041 *	0.021	0.054 **		3.61	0.87

* *p* < 0.05; ** *p* < 0.01; Variables are re-scaled from 0 to 10 to improve comparisons. Source: the authors.

**Table 3 healthcare-09-01388-t003:** Determinants of the job satisfaction of the nurse (total).

Variable	Satisfaction (Coefficients)		
	Model 1	Model 2	Model 3
Intercept	2.76 *	2.03 *	1.91 *
POS.	0.17 *	0.14 *	0.11 *
Empowerment		0.23 *	0.19 *
LMX (m)			0.10 *
*R* ^2^	0.22	0.33	0.36
Δ*R*^2^	0.22	0.11	0.04
*F*	686.32 *	608.11 *	479.06 *
Δ*F*	686.32	415.91	148.90

*N* = 2.500; * *p* < 0.001. The LLX variable does not enter the equation because it is not significant.

**Table 4 healthcare-09-01388-t004:** Correlation of the explanatory variables with the job satisfaction of the nurse.

	Correlation	Statistical Significance
Hospital size	Inverse	<0.001
Beds	Inverse	<0.001
Team size	Inverse	0.003
Span of control	Inverse	0.002
Gender	N.S.	-
Age	N.S.	-
Time working as a nurse	N.S.	-
Time working at hospital	N.S.	-
Time working in unit	N.S.	-
Time with current supervisor	N.S.	-
Full or part-time contract	N.S.	-
University degree	N.S.	-

**Table 5 healthcare-09-01388-t005:** Regression coefficients for the job satisfaction of nurses (by hospital size).

Variable	Satisfaction (Coefficients)					
	Model 1		Model 2		Model 3	
	Small	Large	Small	Large	Small	Large
Intercept	2.62 *	2.17 *	2.42 *	1.97 *	1.99 *	1.87 *
LMX(m)	0.21 *		0.15 *		0.12 *	0.08 *
POS			0.11 *	0.14 *	0.10 *	0.12 *
Empowerment		0.31 *		0.25 *	0.16 *	0.21 *
*R* ^2^	0.20	0.22	0.28	0.35	0.33	0.38
Δ*R*^2^	0.20	0.22	0.08	0.13	0.05	0.03
*F*	229.50 *	456.46 *	177.32 *	423.43 *	148.96 *	323.12 *
Δ*F*	229.50	456.46	100.02	303.63	66.50	80.29

*N_P_* = 906, *N_G_* = 1594 * *p* < 0.001. The supervisor-superior relationship (LLX) is not part of the equations as it was not significant.

**Table 6 healthcare-09-01388-t006:** Regression and statistical coefficients for the moderating variables.

	Small Hospitals		Large Hospitals	
	Coefficient	T-Student	Coefficient	T-Student
Empowerment	0.02	0.81	−0.43	−2.98 *
POS	−0.01	−0.67	−0.01	−0.98

*N_P_* = 906, *N_G_* = 1.594, * *p* < 0.01.

## Data Availability

Due to the nature of this research, participants of this study did not agree for their data to be shared publicly, so supporting data are not available.

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
