# Peer review of "The Quality of the Supervisor–Nurse Relationship and Its Influence on Nurses’ Job Satisfaction"

_healthcare, 2021, doi:10.3390/healthcare9101388_

Round 1
Reviewer 1 Report
It is a very interesting article with an appropiate research design and adequate use of methods. However, I found the discussion too simple and narrow with very little analysis of how these results contribute, support or differ from other similar studies. The fact that there is a lack of a previous theoretical model or developed research hypotheses limits the framework of the discussion. The authors could reinforce the paper´s contribution by enhancing the connections between theory and practice.
Reviewer 2 Report
Overall, I think the manuscript deals with exciting and appropriate issues, i.e., leadership, perceived organisational support, nurses’ work satisfaction, and supervisor-workers interaction. I believe much more empirical research is needed. However, I do think there are some issues in this manuscript that warrant further attention.

Reviewer 3 Report
Thanks for giving me the opportunity of review this paper. Please, see the attached document.

Reviewer 4 Report
Thank you for the opportunity to review this interesting manuscript. I think that the research topic is current in many countries. The manuscript is interesting and useful, but some important information is missing, so I will propose major revisions. I hope that the authors will revise the manuscript.
- Importance: What are the manuscript's strengths? Does it contain new and unique information?
The research topic of this manuscript is important. However, it is not clear for the reader what new information this study brings to the earlier international research literature. That should be highlighted.
Is there a clear statement of the problem and purpose of the study? No, the study objective/aim is not clear, and it needs to be clarified. Objective/aim varies across the text, should be consistent.
No clear research questions have been defined, it would be necessary and reader-friendly to do so and present the research results accordingly.
- Theory: Is the manuscript logical? Is the theory parsimonious?
Does the introduction provide sufficient background and include all relevant references? Main concept needs to be clearly defined.
Does the literature review involve a critical appraisal of the literature rather than a description of the literature? Mainly description of the earlier literature, so critical appraisal should be added, and theoretical background needs to be strengthened.
Does it provide an integrated synopsis of what is known and what is not known about a topic? A limited way, not in the broad sense. This needs to be modified.
- Methodology: Approach? Appropriate design and methods? Is the sample appropriate and adequate? Analysis and interpretation?
Abstract: clear conclusions missing (Conclusions: there seem to be results now)
Keywords: interaction? work or job satisfaction?
Authors do not state how the sample size was calculated, sampling method is also unclear. Sample size needs to be justified.
Is the study context well described? No, study context needs to be described more clearly, and justify. Data collection has been described rather superficially without clear scientific justification.
Is the research design appropriate? Yes, but please clarify the study design
Is the research design stated? No, needs to do so. (e.g. descriptive cross-sectional study)
Is the approach appropriate for the research question? Yes, cross-sectional study and survey are appropriate. Research question(s) needs to be defined.
Are the methods adequately described? No, data collection needs to be described as well as statistical methods used
Is the method of information collection described in enough detail to understand the process? No, data collection needs to be clarified
Was the statistical analysis appropriate for the data? Yes, it is appropriate
Is the method of analysis clearly described? No, all the statistical methods used need to be described in Data analysis section
Is the method of analysis appropriate for the research question? The clear research questions are not defined.
The aim was to confirm > the reader is wondering whether it would have been possible to set hypotheses
Models tested need to be described
- Results
Do the findings answer the research question? Not clearly, there is no clear research questions.
Are the results clearly presented? The presentation of results should be clarified, with reader friendliness in mind
The background information of the respondents is missing, maybe put in the table
- Discussion
Does the research relate the findings to previous work? Yes, but only some extent (should be done in more depth). In the discussion the main results are not logically viewed in terms of research questions, which gives a fragmented picture.
- Adherence to ethical standards:
Is ethical approval for the study and study instruments documented? Yes, but not in all respects.
Has confidentiality been maintained? No found, could be clarified. Informed consent? voluntariness?
Has there been a violation of any accepted norms? Not found
Are appropriate headings and subheadings used to help with the organization of the ideas? Yes, but only partly. Data collection and data analysis, maybe own sections
Quality and clarity of writing
Is the manuscript easy to read? Yes.
References
Are references current? Yes, seem to be
Are key references included? Yes, seem to be
- Other comments
Are the strengths and limitations of the study identified? No, needs to do so. The generalizability of the results should be addressed.
Are areas for further inquiry suggested? Yes.
- Conclusion:
Are the conclusions supported by the results? Clear conclusions are missing
Do the data support the conclusions? Not clearly, conclusions need to be clarified, more far-reaching conclusions should be drawn. Practical implications need to be addressed more clearly.
The reader is not very convinced that the study adds to knowledge in this study area > needs to be highlighted.
I wish all the best for this manuscript.
Round 2
Reviewer 2 Report
Well done and best wishes!
Reviewer 3 Report
It must be recognized that the authors have incorporated changes that have notably improved the work. Congratulations.
Reviewer 4 Report
Many thanks for these modifications, good work.
I have no further comments.